# The functioning of different beetle (Coleoptera) sampling methods across altitudinal gradients in Peninsular Malaysia

**Muneeb M. Musthafa**[1,2,3]*, **Fauziah Abdullah**[2,3,4], **Matti J. Koivula**[5]

**1** Department of Biosystems Technology, Faculty of Technology, South Eastern University of Sri Lanka, University Park, Oluvil, Sri Lanka, **2** Institute of Biological Science, Faculty of Science, University of Malaya, Kuala Lumpur, Malaysia, **3** B513, Toxicology Lab, Institute of Postgraduate Studies, University Malaya, Kuala Lumpur, Malaysia, **4** Center of Biotechnology in Agriculture, University Malaya, Kuala Lumpur, Malaysia, **5** Natural Resources Institute Finland (LUKE), Helsinki, Finland

* muneeb@seu.ac.lk

**Data Availability Statement:** All relevant data are available within the paper and Supporting information.

## Abstract

Biodiversity research relies largely on knowledge about species responses to environmental gradients, assessed using some commonly applied sampling method. However, the consistency of detected responses using different sampling methods, and thus the generality of findings, has seldom been assessed in tropical ecosystems. Hence, we studied the response consistency and indicator functioning of beetle assemblages in altitudinal gradients from two mountains in Malaysia, using Malaise, light, and pitfall traps. The data were analyzed using generalized linear mixed-effects models (GLMM), non-metric multidimensional scaling (NMDS), multivariate regression trees (MRT), and indicator species analysis (IndVal). We collected 198 morpho-species of beetles representing 32 families, with a total number of 3,052 individual beetles. The richness measures generally declined with increasing altitude. The mountains differed little in terms of light and Malaise trap data but differed remarkably in pitfall-trap data. Only light traps (but not the other trap types) distinguished high from middle or low altitudes in terms of beetle richness and assemblage composition. The lower altitudes hosted about twice as many indicators as middle or high altitudes, and many species were trap-type specific in our data. These results suggest that the three sampling methods reflected the altitudinal gradient in different ways and the detection of community variation in the environment thus depends on the chosen sampling method. However, also the analytical approach appeared important, further underlining the need to use multiple methods in environmental assessments.

## Introduction

Most hotspots for global biodiversity can be found in tropical regions [1] which are thus of central importance for conservation. Tropical species often have limited distributions, particularly those species that occupy higher altitudes at mountain slopes [2]. Such mountain species are often endemic to these regions, and also sometimes such that may not be able to move

**Funding:** We thank the University of Malaya for financing this study (grant numbers RP004E/13SUS and PG059/2014B). The amended Role of Funder: The University of Malaya funded by two grants namely, RP004E/13SUS and PG059/2014B mainly for field works, travelling and sample collection.

elsewhere if conditions turn unfavorable for them due to, for example, habitat destruction or climate change [3, 4]. Hence, studies on mountain species communities provide crucial information for global conservation efforts, in addition to these environments being particularly vulnerable to habitat degradation and alterations in landscape structure [5, 6]. Our research contributes to the knowledge on biodiversity in Malaysian mountain forests, a little studied biome.

Community composition is largely determined by the relative abundances of and interactions among its species, which contribute to the general biodiversity response to environmental variation. Communities vary according to, among other aspects, latitude and altitude [7–9]. Our understanding regarding this variation relies on some commonly assessed taxonomic groups and associated sampling methods. Similarly, land-use planning or conservation decisions may be based on only a handful of well-known species or a single sampling method. Insects, for instance, are often sampled using light, Malaise, or pitfall traps (e.g., [10]). Environmental assessments, however, are often based on just one sampling method, which is assumed to produce a general biodiversity response, but this assumption has seldom been challenged.

Earlier research indicates that the three mentioned trap types differ in their efficiency in capturing specimens and species (e.g., [11, 12] though the efficiency could vary with the composition of the local species community determined by, for example, altitude or habitat type. As different methods are known to capture partly different species (e.g., [13]), their use may, at least theoretically, result in different conclusions and hence applications in land use or conservation.

The objective of the present study is to evaluate the consistency of light, Malaise and pitfall trap samples in reflecting an altitudinal gradient in tropical Malaysia. More specifically, we compare the three sampling methods across the altitudinal gradient from four viewpoints:

1. Assemblage species richness;

2. Assemblage species-compositional turnover;

3. Assemblage composition; and

4. Species associated with different combinations of sampling method and altitude

## Materials and methods

### Study sites

The Titiwangsa mountain range dominates the landscape of Peninsular Malaysia. Within this range, we sampled beetles at Fraser's Hill (3˚43' N, 101˚45' E) and Genting Highland (3˚25' N, 101˚47' E) that are 95 km apart (Fig 1). Fraser's Hill mountain tops peak at between 1,000 and 1,800 m a.s.l., whereas the Genting Highland peaks at about 1,800 m a.s.l. In this region, wet and dry seasons cannot be differentiated, as the annual rainfall of 1,800–3,500 mm is distributed throughout the year [14, 15]. Temperature, humidity and luminosity at our study sites, collected for another manuscript by the author MMM, correlated weakly with altitude; correlation coefficients were -0.12 (p = 0.296), -0.27 (p = 0.022) and 0.27 (p = 0.0219) for temperature, humidity and luminosity, respectively.–Collection permits for Fraser's Hill and Genting Highlands were granted by the Forest Department of Malaysia.

Fraser's Hill, locally known as Bukit Fraser, is a well-preserved permanently-protected nature reserve located at the Raub district of Pahang state. Fraser's Hill has been developed as a hill station dating back in 1919 [16], where 90% of 2,800-ha land area is covered by forests

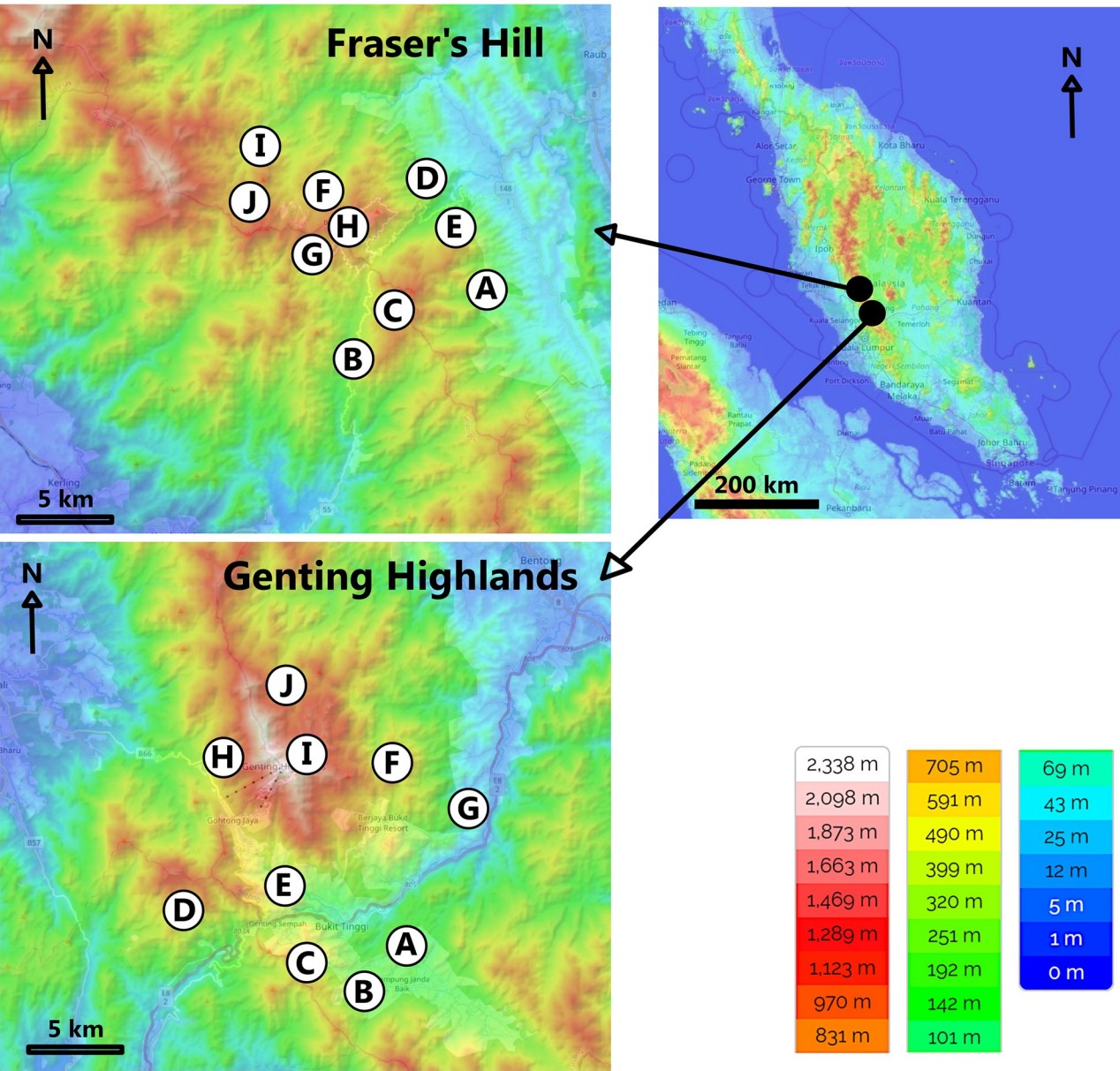

**Fig 1. Study location in Malaysia.** Right: map of Malaysia, with altitudes indicated with different colors, from blue (low) to green, yellow, orange and red (high). Left-hand graphs show locations of ten study stands in each study mountain (denoted with letters A-J; compare Table 1); Altitudes are indicated with colors, as in the overall Malaysian map. Map copyright: free open-source maps at https://en-gb.topographic-map.com/legal/; altitudinal data from [69]. We have modified the original maps by adding compass arrows, scale bars and stand codes.

[14, 17]. Fraser's Hill comprises residential areas, commercial complexes, community services, and recreational facilities. The forests mainly consist of tropical montane cloud forests. Lower montane forests–at about 500–1,200 m a.s.l.–are dominated by Fagaceae and Lauraceae trees, whereas upper montane forests–from about 1,200 to 1,800 m a.s.l.–are dominated by Coniferae, Ericaceae, and Myrtaceae trees [18].

Genting Highland is perhaps the most disturbed cloud forest in the Malaysian mountains. The summit is covered by amusement parks, casinos, and hotels [19]. The highest peaks at Genting Highland reach 1,800 m a.s.l, where 96% of the total of 3,965 ha of land is still covered

with mostly primary forests [14]. Before Genting Highland became an entertainment site, it was an undisturbed forest that could be reached only via jungle trekking [20, 21].

## Sampling methods

We sampled beetles using two Malaise traps, two light traps, and 10 (5 groups of 2) pitfall traps at each altitude (500 m, 1,000 m, 1,500 m, and 1,800 m a.s.l.) at both mountain slopes. All the traps (or trap groups for pitfall traps) were set at least 150 m apart at a given altitude and were placed at least 150 m from the nearest main road. These roads and a river split the sampled forests into a total of twenty distinctive stands; each of these had at least one type of trap (Fig 1, Table 1).

Malaise traps consisted of a nylon net connected to a collection jar, half filled with 70% ethanol and attached to a tree branch about one meter above the ground. Light traps had a mosquito net with attached 160-watt mercury bulb connected to a portable Honda EU10i generator. Pitfall traps were transparent, colourless plastic cups (diameter 65 mm, depth 95 mm) partly filled with 70% ethanol and dug into the ground with the rim flush with the soil surface. We placed large dry leaves above each pitfall trap to protect them from litter and rain.

We sampled beetles once per month in October 2014, and March, June, and September 2015. At each collecting date, Malaise and pitfall traps were set for 24 hours, starting at 08:00 AM, and light traps operated from 18:00 to 23:30. In the latter, beetles were obtained manually from the traps using collection bottles and aspirators. We occasionally continued to use the light traps until the next morning at 06:00 AM but did not capture additional beetles.

**Table 1. Sampling effort at Fraser's Hill (FH) and Genting Highland (GH) mountains.**

| Mountain | Altitude | Stand ID | Light | Malaise | Pitfall |
|----------|----------|----------|-------|---------|---------|
| FH | 500 | A | - | - | 1 |
| FH | 500 | B | - | - | 4 |
| FH | 500 | C | 1 | - | - |
| FH | 1,000 | D | 1 | 2 | 5 |
| FH | 1,000 | E | 1 | - | - |
| FH | 1,500 | F | 1 | - | 2 |
| FH | 1,500 | G | - | 1 | 1 |
| FH | 1,500 | H | 1 | 1 | 2 |
| FH | 1,800 | I | 1 | 2 | 1 |
| FH | 1,800 | J | 1 | - | 4 |
| GH | 500 | A | - | 1 | - |
| GH | 500 | B | - | 1 | - |
| GH | 500 | C | 2 | - | 5 |
| GH | 1,000 | D | - | - | 2 |
| GH | 1,000 | E | 2 | 2 | 3 |
| GH | 1,500 | F | - | 2 | 4 |
| GH | 1,500 | G | - | - | 1 |
| GH | 1,500 | H | 2 | - | - |
| GH | 1,800 | I | 2 | 1 | 4 |
| GH | 1,800 | J | - | 1 | 1 |

In both mountains, traps were set at four Altitudes: 500, 1,000, 1,500 and 1,800 m a.s.l., in ten stands in both mountains; stands are indicated with letters A-J (compare Fig 1). The three right-hand columns show the number of traps (or groups of five pitfall traps) in each stand.

In statistical analyses, we pooled the four periods for each trap (or trap group for pitfall traps) and consider each trap (or trap group for pitfall traps) as a replicate (initially 72; 4 altitudes and 2 mountains, each with 2 light and 2 Malaise, and 5 groups of pitfall traps).

## Identification of specimens

We sorted, counted, and cross-checked all beetle specimens using available keys [22–36]. We confirmed the identification of difficult specimens at the collections of the Wildlife Department of Malaysia, University of Malaya, National University of Malaysia, and Forestry Department of Malaysia. The identification of beetles to species is hampered by the lack of experts and species compilations. In the present study, the samples contained at least five species new to science, which will be described in later papers. Clearly, the often broad taxonomic levels and the shortage of knowledge about the ecological traits of species must be acknowledged while interpreting results, as similar-looking species might be different in terms of, for example, life cycles, diets, abilities to disperse, and habitat requirements.

## Statistical analysis

We had initially 72 samples (16 for light, 16 for Malaise, and 40 for pitfall traps [5 sets * 2 * 4 altitudes}). Three samples, however, produced only 1–2 species and were excluded due to the difficulty in calculating pair-wise dissimilarities and richness estimates (one light and two Malaise trap samples). Thus, we ran all analyses with 69 samples (Table 1). We ran the following analyses for the four viewpoints proposed above.

## Assemblage richness

To assess sampling-method dependent variation in species richness, we subjected the beetle data to a generalized linear mixed-effects model (GLMM; [37] to quantify variation according to altitude (500, 1,000, 1,500, or 1,800 m a.s.l.). We ran the model separately for each sampling method (light, Malaise, or pitfall traps; hereafter "Method" for brevity) to be able to compare their similarity in responses to altitude. We subjected the observed number of species, richness estimates based on rarefaction standardization to 5, 10, and 20 individuals, and asymptotic values of coverage-based rarefaction estimates (hereafter coverage-based asymptotic richness; [38, 39] to a GLMM with Altitude as a fixed factor and Mountain (Fraser's Hill or Genting Highland; Fig 1, Table 1) as a random factor (to account for spatial autocorrelation), using lme4 [40]. For full model outputs, see S1 Table.

For linear models it is important to assess the normality and homoskedasticity of model residuals, which we did in two ways. Firstly, we inspected Q-Q plots of GLMM residuals visually (car package; [41], confirmed by Wilk-Shapiro test for residual normality (S1 Fig and S2 Table). These did not indicate major departures from normality, except in the pitfall-trap data. Secondly, we ran Wilk-Shapiro test to check the normality, and Breusch-Pagan test to assess homoskedasticity, of ANOVA residuals (model Mountain + Altitude; S3 Table). These checks indicated the following issues: (i) heteroskedasticity in the number of species and non-normality in the rarefied richness to 5 individuals in the light-trap data; (ii) heteroskedasticity in the number of species in the Malaise-trap data; and (iii) non-normality in the rarefied richness standardized to 5 and 10 individuals and coverage-based asymptotic richness in the pitfall-trap data (as for GLMM residuals above). Due to these deviations from normality, we reran the GLMM models (as in Table 2) using Robust LMM (robustlmm package; [42]. Models run using biological data often contain outlier samples that may render residual distributions non-normal or heteroskedastic. Estimates from Robust LMM are little affected by such outliers, if the tuning parameter (k) is set at a low value; values approaching ∞ produce results similar to

**Table 2. GLMM summary for altitude, using mountain as a random factor, for the number of species and rarefaction standardized number of species for 5, 10 or 20 individuals.**

| Variable/Category | LIGHT TRAPS | | MALAISE TRAPS | | PITFALL TRAPS | |
|---|---|---|---|---|---|---|
| | %var | Effect | %var | Effect | %var | Effect |
| *Number of species* | | | | | | |
| Mountain | 7 | | 4 | | 53 | |
| Altitude | 41 | | 78 | | 11 | |
| * 1,000 m a.s.l. | | ns | | neg | | (pos) |
| * 1,500 m a.s.l. | | ns | | neg | | ns |
| * 1,800 m a.s.l. | | neg | | neg | | ns |
| Residuals | 52 | | 19 | | 35 | |
| *Rarefied richness to 5 individuals* | | | | | | |
| Mountain | 7 | | 6 | | 17 | |
| Altitude | 33 | | 70 | | 16 | |
| * 1,000 m a.s.l. | | ns | | ns | | pos |
| * 1,500 m a.s.l. | | ns | | ns | | ns |
| * 1,800 m a.s.l. | | (neg) | | neg | | ns |
| Residuals | 60 | | 24 | | 67 | |
| *Rarefied richness to 10 individuals* | | | | | | |
| Mountain | 15 | | 7 | | 26 | |
| Altitude | 45 | | 72 | | 11 | |
| * 1,000 m a.s.l. | | ns | | ns | | pos |
| * 1,500 m a.s.l. | | ns | | ns | | ns |
| * 1,800 m a.s.l. | | neg | | neg | | ns |
| Residuals | 40 | | 21 | | 63 | |
| *Rarefied richness to 20 individuals* | | | | | | |
| Mountain | 4 | | 7 | | 28 | |
| Altitude | 61 | | 72 | | 12 | |
| * 1,000 m a.s.l. | | ns | | (neg) | | pos |
| * 1,500 m a.s.l. | | ns | | neg | | ns |
| * 1,800 m a.s.l. | | neg | | neg | | ns |
| Residuals | 35 | | 21 | | 59 | |
| *Coverage-based asymptotic richness* | | | | | | |
| Mountain | 17 | | 25 | | 35 | |
| Altitude | 41 | | 57 | | 3 | |
| * 1,000 m a.s.l. | | ns | | neg | | ns |
| * 1,500 m a.s.l. | | ns | | neg | | ns |
| * 1,800 m a.s.l. | | neg | | (neg) | | ns |
| Residuals | 42 | | 18 | | 61 | |

Results are shown for light, Malaise and pitfall trap data. Numbers in "%var" columns are percentages explained by a given variable; letters in"Effect" columns show whether a given Altitude differed significantly (p < 0.05) and positively (pos) or negatively (neg) from the lowest Altitude (500 m a.s.l.) (in parentheses if marginally significant, i.e., p < 0.1) or whether this comparison was non-significant (ns; p >0.1). For full output, see S1 Table.

a normal GLMM. Here, we applied the smoothed Huber (k = 1.345, s = 10) function for fitting random effects (Mountain) and variance component (Altitude), as recommended by [42] (S4 Table). As the relative magnitudes and directions of estimates in Robust LMM were very similar to the initial GLMM (with the exception of coverage-based asymptotic richness for pitfall traps), we conclude that our GLMM results shown in Table 2 are robust.

## Assemblage turnover

To examine possible species turnover according to altitude and Method, we calculated averages and standard errors for 20 most abundant species, separately for each Altitude and Method. We plotted these values according to Altitude, using the rank order of the abundances of species captured using each Method. We evaluated the community turnover, or distinctiveness of samples, between Altitudes using permutational multivariate ANOVA (the adonis function in R package vegan, with Mountain [Fraser's Hill or Genting Highland] as strata; [43]. Here, we only considered single or combinations of subsequent Altitudes. Thus, for example, a combination 500 + 1000 m was considered but not 500 + 1800 m. As a simple measure of turnover between Methods, we indicated in these plots species that were unique for a given Method, and those species that were shared with 1–2 other Methods. We confirmed this comparison of Methods using permutational multivariate ANOVA, as described above.

## Assemblage composition

To examine beetle assemblage composition across Altitudes and Methods, we used two analyses. Firstly, we used non-metric multidimensional scaling (NMDS; [44]) to assess variation in community composition, using the vegan package [43]. We used Method-specific data sets by applying a Bray-Curtis dissimilarity matrix. We used the above-described permutational multivariate ANOVA for Altitudes as a confirmation of the NMDS result. Secondly, we subjected the Method-specific beetle data sets to multivariate regression tree analysis (MRT; [45]) based on the Bray-Curtis dissimilarity matrix, using the mvpart package [46]. We used altitude and, as our earlier research has indicated differences in beetle faunas between Fraser's Hill and Genting Highland [47], Mountain as explanatory variables. MRT identifies groups of samples as defined by explanatory variables (Mountain and Altitude) and is not restricted by non-linearities [45]. We present the result as a tree of dichotomies, where each dichotomy is based on minimizing the dissimilarity of samples within each tree branch. We report the tree with the lowest cross-validated relative error, following the 1-SE rule [48]. Cross-validated relative error provides a better estimate than relative error for the predictive accuracy of the MRT for a new dataset [45].

## Species associations with altitude and method

To detect species characteristic to particular combinations of Method and Altitude, we calculated an indicator value (IndVal; [49, 50] for each species, based on all logical combinations of Method and Altitude. Here, we used the indicspecies package [51] and allowed each Method to appear singly or jointly with 1–2 other Methods, whereas for Altitude we only considered single, or combinations of subsequent, Altitudes, as described above. We restricted the IndVal to species with a total sample of at least five individuals.

# Results

## Beetle richness according to mountain and altitude

We collected 198 morpho-species of beetles representing 32 families with a total number of 3052 individuals (S5 Table). Nine taxa were identified to species, 143 to genus, and 43 to higher taxonomic levels. We refer to all these as "species" below for convenience. Regarding Method, we collected 107 species using light traps, 127 using Malaise, and 45 using pitfall traps. A total of 135 species were represented by only one Method, whereas 45 had been captured with two and 18 with all three sampling methods. Altogether 98 species were singletons or doubletons, whereas 84 were represented by at least five individuals (S5 Table). Species accumulation

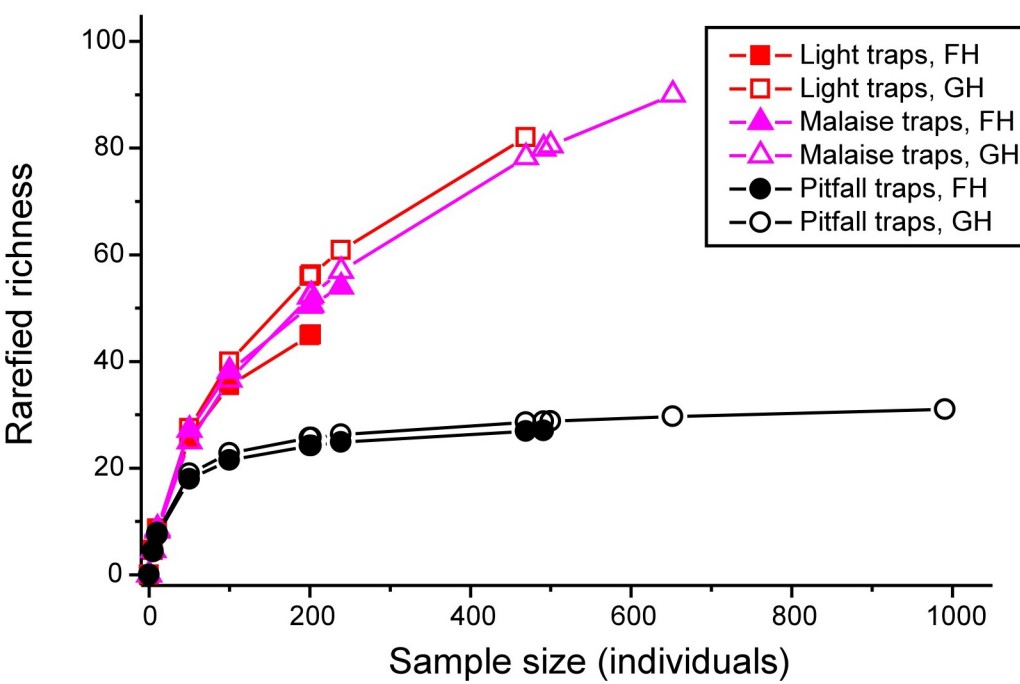

**Fig 2. Rarefaction curves for the three trap types and the pooled sample.** The end point of each curve indicates the trap-type specific or pooled (total) number of individuals.

curves based on rarefaction suggested that pitfall traps had captured nearly all potential species at about 200 individuals at both Mountains, whereas the accumulation curves for light and Malaise traps were still very steep at about 400–500 individuals (Fig 2).

The GLMM performed rather similarly in terms of the four measures of richness, but the three Methods produced partly different results (Table 2, Fig 3; S1 Table). The Mountains differed little in terms of light and Malaise trap data but remarkably in pitfall-trap data. The richness measures based on light traps declined with Altitude but only 1,800 m differed significantly from 500 m. The decline with Altitude occurred also in Malaise traps so that all higher Altitudes differed significantly from 500 m. Richness measures based on pitfall traps, on the other hand, peaked at 1,000 m (Table 2, Fig 3).

## Species turnover

The rank-abundance plots of twenty most abundant species (Fig 4) reflected relatively high similarity between light and Malaise traps, with nine of the 20 most numerous species being shared (grey columns), whereas only one of the 20 dominant species–*Pityogenes* sp1 –in pitfall-trap samples was shared with light- and Malaise-trap samples (white columns). For species identities in this graph, see S6 Table. Permutational multivariate ANOVA indicated that also light and Malaise traps were compositionally different; for light vs. Malaise traps, $F = 3.21$, $R2 = 0.11$, $p = 0.0010$; for light vs. pitfall traps $F = 9.41$, $R2 = 0.15$, $p = 0.0010$; and for Malaise vs. pitfall traps $F = 10.48$, $R2 = 0.17$, $p = 0.0010$. Moreover, the plots suggested more abundance or occurrence changes with Altitude in Malaise and light traps, whereas the plots varied less and more erratically in pitfall-trap samples (Fig 4). Another striking pattern in light-trap samples (Fig 4, left) was the lack of the 12 most abundant species at 1,800 m. Only 9% of light-trap species and 17% of Malaise-trap species were found in both 500 m and 1,800 m, whereas the percentage was 63 for pitfall-trap samples.

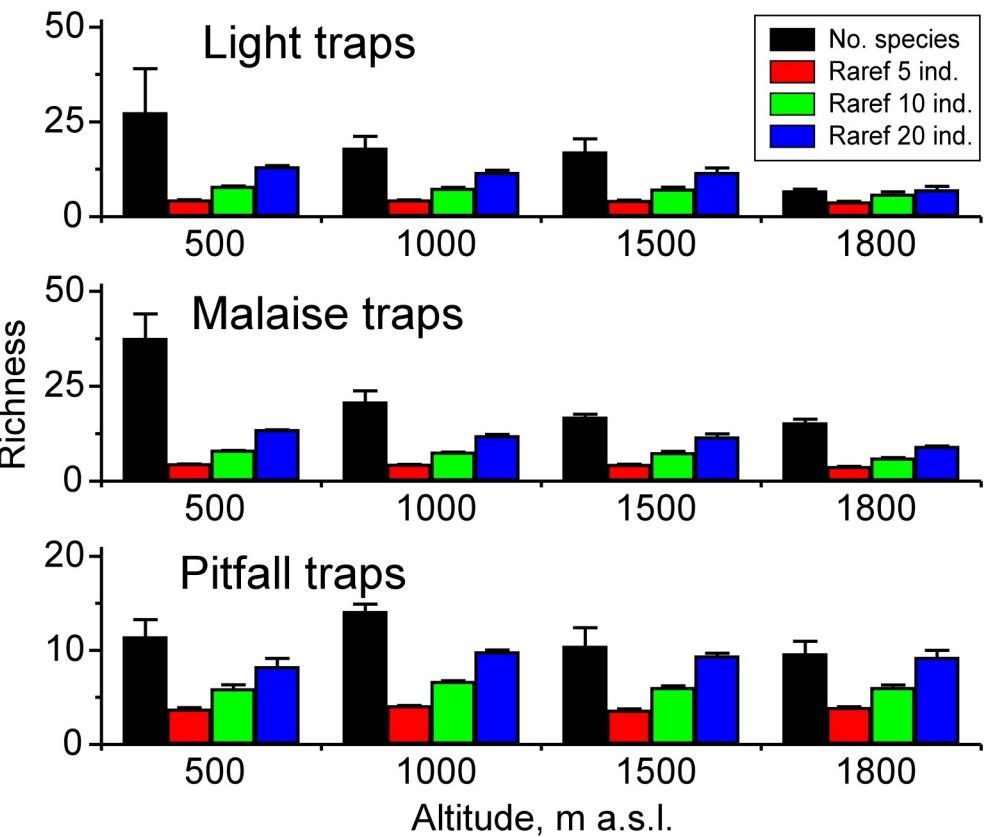

**Fig 3. Average samples + SE of four measures of species richness for three sampling methods according to altitude.** Number of species, and rarefaction standardized richness based on 5, 10 and 20 individuals are shown. Note that the X axis length for pitfall trap samples is different.

Permutational multivariate ANOVA confirmed compositional changes with Altitude, particularly at 1,800 m, but the three Methods showed different patterns in this respect (Table 3). In light traps, only 1,800 m differed significantly from the other Altitudes; however, all except one combination of 2–3 Altitudes also differed significantly from the rest of the Altitudes. The patterns were similar for Malaise traps, except that different combinations of 2–3 Altitudes differed less commonly from the rest of the Altitudes. Regarding pitfall traps, then, all Altitudes and their combinations (except for 1,000–1,500 m) had distinctive assemblages (Table 3).

## Assemblage composition according to Method and Altitude

According to NMDS, samples in light and Malaise traps distinguished 1,800 m from the three lower altitudes, whereas pitfall trap samples formed two sample clusters, of which only the other distinguished clearly between Altitudes (Fig 6). Thus, in pitfall-trap samples, Altitude was reflected in Fraser's Hill but not in Genting Highland (on the left and right, respectively, in Fig 5).

MRT for light-trap data consistently resulted in a one-dichotomy tree (Fig 6). The only division, where 1,800 m was split from the rest of the Altitudes, explained 47% of the variation in light-trap data, the former being species poor as is evident from rank-abundance plots in the end branches of MRT (Fig 6). Malaise-trap data, on the other hand, suggested that the same dichotomy, based on Altitude, was apparent in Fraser's Hill but not in Genting Highland;

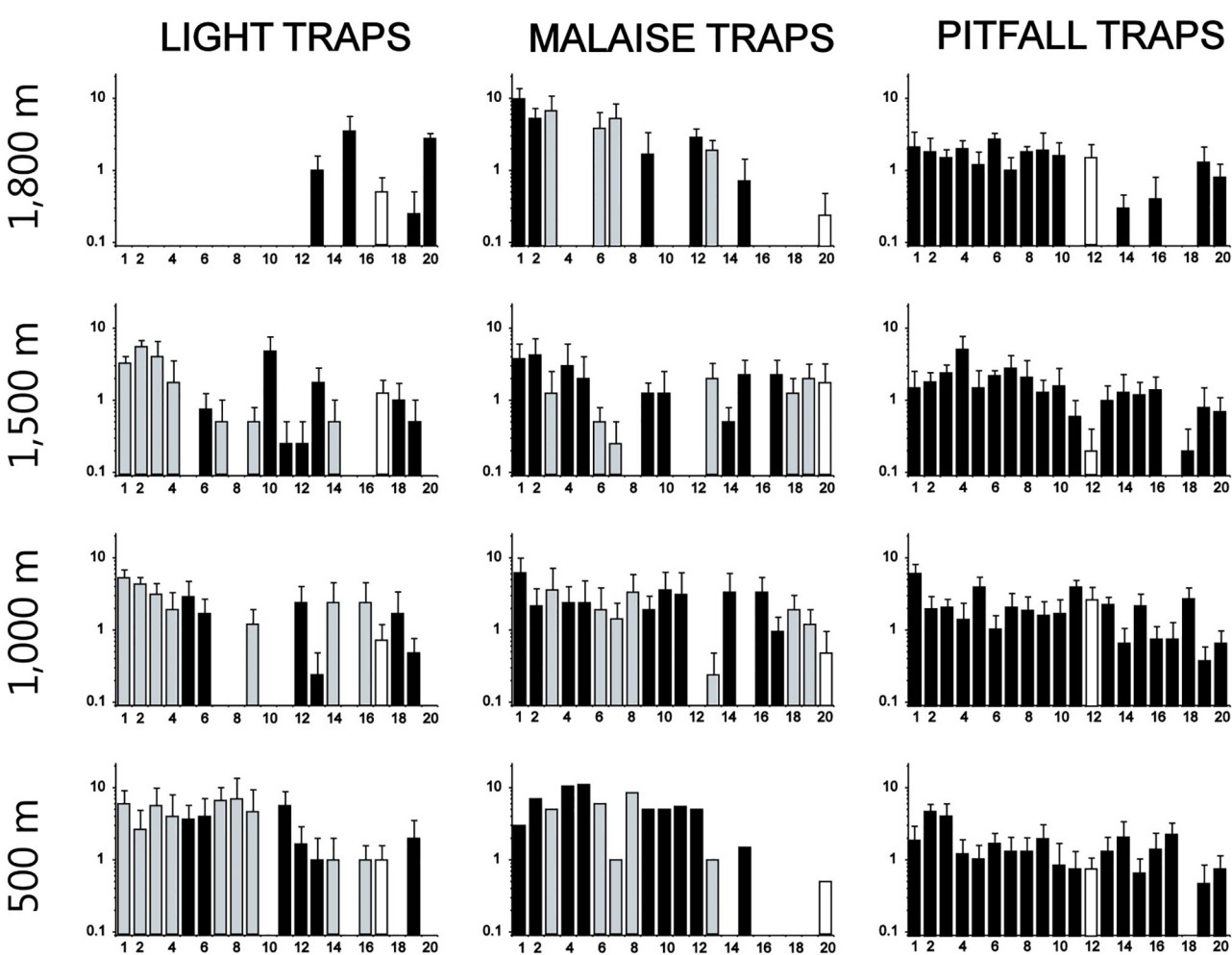

**Fig 4. Rank abundance (mean + SE) of twenty most abundant species in each trap type at different altitudes (500, 1,000, 1,500 and 1,800 m a.s.l.).**
Note that species are mostly different for each trap type, each sorted according to the rank order of the total number of individuals in that trap type.
Grey columns show species that were shared between light and Malaise traps, and white columns show the shared species between pitfall, light and
pitfall traps (one species). For species identities, see S5 Table.

again, 1,800 m had fewer species though this was based on only two samples. This two-dichotomy tree explained 81% of the variation in the Malaise trap data (Fig 6). Also, pitfall trap data suggested that Altitudes could be distinguished at Fraser's Hill but not at Genting highland; here, 1,500 and 1,800 m diverged from the two lower Altitudes that, in turn, were split in the third dichotomy (Fig 6).

## Species associated with different combinations of Method and Altitude

We found 83 indicator species for different combinations of Method and Altitude (Table 4). Thirteen species were commonly caught with two sampling methods (12 for light and Malaise traps, and one with Malaise and pitfall traps). Moreover, eight species indicated pitfall traps across the full altitudinal range (500–1,800 m), whereas we did not detect such altitude-independent indicators for the two other sampling methods. Light traps produced 12 indicators of low (500–1,000 m), six of the middle (1,000–1,500 m) and seven of high (1,500–1,800 m) Altitudes, whereas the respective numbers were 16, four, and six for Malaise and ten, eleven and

**Table 3. Permutational multivariate ANOVA to assess the distinctiveness of beetle communities at different altitudes (Alt 500, 1,000, 1,500 or 1,800 m a.s.l., or their logical combinations) as reflected by using three sampling methods (compare Fig 2).**

| | LIGHT TRAPS | | | | | | MALAISE | | | | | | PITFALL | | | | | |
|---|---|---|---|---|---|---|---|---|---|---|---|---|---|---|---|---|---|---|
| Variable | df | SS | MS | F | R² | p | df | SS | MS | F | R² | p | df | SS | MS | F | R² | p |
| Alt 500 | 1 | 0.47 | 0.47 | 1.28 | 0.09 | 0.0659 | 1 | 0.40 | 0.40 | 1.17 | 0.09 | 0.2717 | 1 | 0.48 | 0.48 | 1.58 | 0.04 | 0.0300 |
| Residuals | 13 | 4.75 | 0.37 | 0.91 | | | 12 | 4.12 | 0.34 | 0.91 | | | 38 | 11.55 | 0.30 | 0.96 | | |
| Alt 1000 | 1 | 0.47 | 0.47 | 1.29 | 0.09 | 0.1548 | 1 | 0.42 | 0.42 | 1.23 | 0.09 | 0.1528 | 1 | 0.97 | 0.97 | 3.34 | 0.08 | 0.0010 |
| Residuals | 13 | 4.75 | 0.37 | 0.91 | | | 12 | 4.10 | 0.34 | 0.91 | | | 38 | 11.06 | 0.29 | 0.92 | | |
| Alt 1500 | 1 | 0.53 | 0.53 | 1.47 | 0.10 | 0.0939 | 1 | 0.42 | 0.42 | 1.24 | 0.09 | 0.1299 | 1 | 0.46 | 0.46 | 1.51 | 0.04 | 0.0390 |
| Residuals | 13 | 4.69 | 0.36 | 0.90 | | | 12 | 4.10 | 0.34 | 0.91 | | | 38 | 11.57 | 0.30 | 0.96 | | |
| Total | 14 | 5.22 | 1.00 | | | | 13 | 4.52 | 1.00 | | | | 39 | 12.03 | 1.00 | | | |
| Alt 1800 | 1 | 1.36 | 1.36 | 4.56 | 0.26 | 0.0020 | 1 | 0.70 | 0.70 | 2.21 | 0.16 | 0.0030 | 1 | 0.76 | 0.76 | 2.56 | 0.06 | 0.0020 |
| Residuals | 13 | 3.87 | 0.30 | 0.74 | | | 12 | 3.82 | 0.32 | 0.84 | | | 38 | 11.27 | 0.30 | 0.94 | | |
| Alt 500–1000 | 1 | 0.67 | 0.67 | 1.93 | 0.13 | 0.0290 | 1 | 0.47 | 0.47 | 1.40 | 0.10 | 0.0699 | 1 | 0.97 | 0.97 | 3.33 | 0.08 | 0.0010 |
| Residuals | 13 | 4.55 | 0.35 | 0.87 | | | 12 | 4.05 | 0.34 | 0.90 | | | 38 | 11.06 | 0.29 | 0.92 | | |
| Alt 1000–1500 | 1 | 0.98 | 0.98 | 3.01 | 0.19 | 0.0010 | 1 | 0.68 | 0.68 | 2.13 | 0.15 | 0.0420 | 1 | 0.55 | 0.55 | 1.82 | 0.05 | 0.0070 |
| Residuals | 13 | 4.24 | 0.33 | 0.81 | | | 12 | 3.84 | 0.32 | 0.85 | | | 38 | 11.48 | 0.30 | 0.95 | | |
| Alt 1500–1800 | 1 | 0.67 | 0.67 | 1.93 | 0.13 | 0.0380 | 1 | 0.47 | 0.47 | 1.40 | 0.10 | 0.0619 | 1 | 0.97 | 0.97 | 3.33 | 0.08 | 0.0010 |
| Residuals | 13 | 4.55 | 0.35 | 0.87 | | | 12 | 4.05 | 0.34 | 0.90 | | | 38 | 11.06 | 0.29 | 0.92 | | |
| Alt 500–1500 | 1 | 1.36 | 1.36 | 4.56 | 0.26 | 0.0020 | 1 | 0.70 | 0.70 | 2.21 | 0.16 | 0.0050 | 1 | 0.76 | 0.76 | 2.56 | 0.06 | 0.0020 |
| Residuals | 13 | 3.87 | 0.30 | 0.74 | | | 12 | 3.82 | 0.32 | 0.84 | | | 38 | 11.27 | 0.30 | 0.94 | | |
| Alt 1000–1800 | 1 | 0.47 | 0.47 | 1.28 | 0.09 | 0.0699 | 1 | 0.40 | 0.40 | 1.17 | 0.09 | 0.2408 | 1 | 0.48 | 0.48 | 1.58 | 0.04 | 0.0210 |
| Residuals | 13 | 4.75 | 0.37 | 0.91 | | | 12 | 4.12 | 0.34 | 0.91 | | | 38 | 11.55 | 0.30 | 0.96 | | |

five for pitfall traps (Table 4). Moreover, three species occurred in pitfall traps across a wider Altitudinal range: two species were found at 500–1,500 m and one at 1,000–1,800 m.

## Discussion

### Beetle species turnover between methods and across the altitudinal gradient

We found that light and Malaise trap samples shared about half of the abundant species, whereas pitfall trap samples were distinctive in this respect. Light and Malaise traps may have attracted a shared pool of species that flies actively, whereas pitfall traps capture mostly ground dwellers that seldom fall into the two other types of traps (e.g., [13]). This finding suggests that, at least in the study region, the variation in beetle communities caused by altitude or associated climatic factors may be better captured if pitfall traps are used together with either light or Malaise traps, whereas a combination of light and Malaise traps may not be equally efficient.

At 1,800 m light (but not Malaise or pitfall) trap samples lost all 12 most abundant species of the total light-trap sample. This loss might have resulted from the captured species pool being more sensitive to altitude-associated changes in abiotic or biotic conditions. However, some of the 12 "lost" species were still captured at 1,800 m using Malaise traps, so another explanation may be trap functioning. Perhaps light or wind conditions–or some other factors related to, for example, vegetation or moisture–were different at the highest altitudes, which might, in turn, have impacted the visibility of light traps or flight activity of many species [52–56]. It would be important to continue beetle monitoring at these mountain tops to see whether the altitudinal distributions of species indeed change with climate and whether the currently dominant mountain-top species persist or disappear.

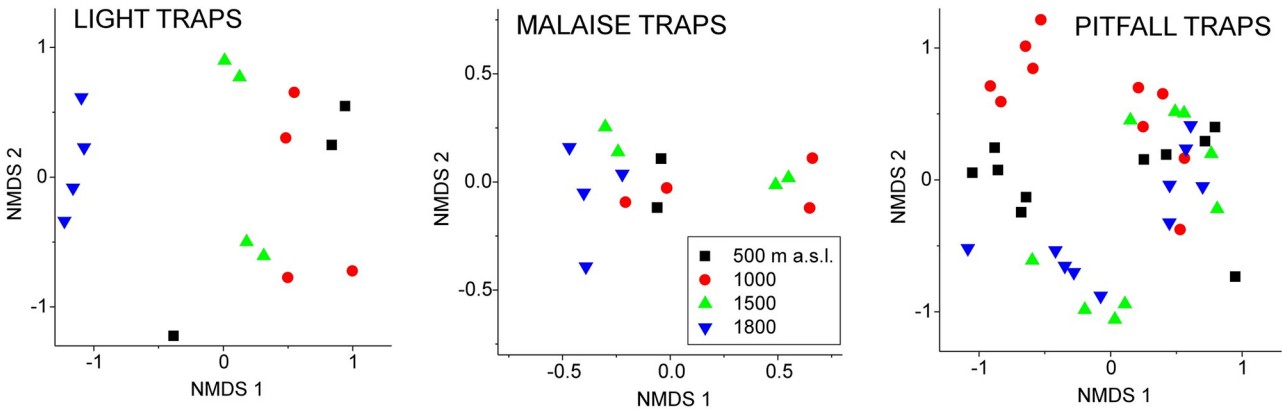

**Fig 5. NMDS plots for Malaysian beetle data, separate runs for three sampling methods (light, Malaise or pitfall traps) at different altitudes (500 m = black squares, 1,000 m = red circles, 1,500 m = green up-triangles and 1,800 m = blue down-triangles).** For statistical comparisons between Altitudes, see Table 3.

The relative similarity of rank-abundance plots at different altitudes, particularly for pitfall-trap samples, may have occurred because many dominant species are adapted to a wide range of altitudes (e.g., [57] and/or temperature or moisture conditions [58, 59]. Pitfall traps capture mostly ground dwellers, and field-layer vegetation or soil conditions may thus have kept conditions relatively constant across the altitudinal gradient for these species (see also Materials and methods). These factors might decrease variation in micro-climate [60], perhaps through the sheltering effect of forest trees [61].

## Beetle community structure across the altitudinal gradient using different methods

Our community analyses suggest that the overall beetle community varied remarkably according to altitude, but the magnitude of this response depended on geographic location (the two mountains) and captured species subset (sampling method). Thus, according to NMDS and MRT, light traps distinguished 1,800 m from lower altitudes, whereas Malaise and pitfall traps reflected primarily differences between the two mountains and only secondarily altitudinal

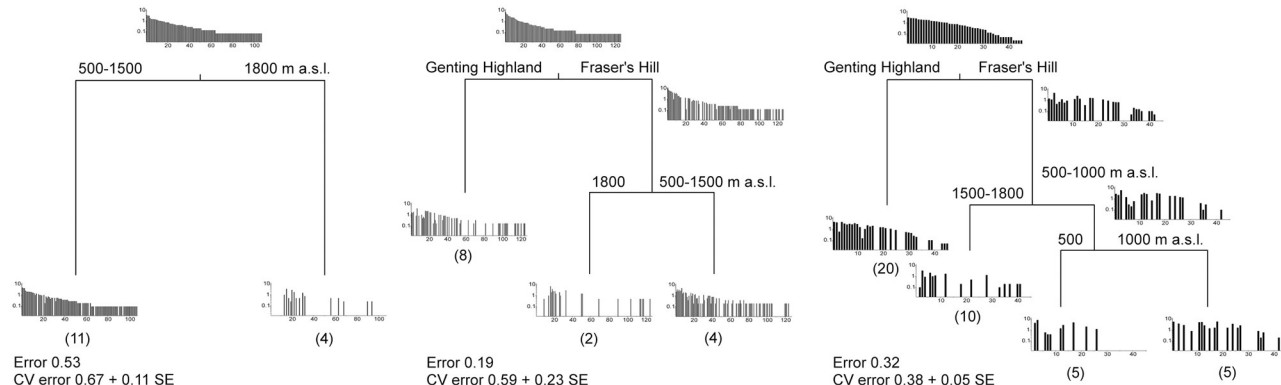

**Fig 6. MRT for sampling-method specific beetle data, using mountain (Fraser's Hill or Genting Highland) and altitude (500, 1,000, 1,500 or 1,800 m a.s.l.) as explanatory variables.** The column plots show the relative abundance of each species captured using a given trap type, sorted according to the rank order of abundance of the total trap-type specific sample. Numbers in parentheses below each end branch show the number of samples falling into that branch.

**Table 4. Significant (p < 0.05) indicators with n > 4 in the beetle data.**

| Category | Species | IndVal | Category | Species | IndVal |
|---|---|---|---|---|---|
| **Single-method indicators** | | | **Single-method indicators continued** | | |
| **LT 500** | *Anomala* sp2 | 0.81 | PT | *Harpalus* sp1 | 0.84 |
| **LT 500** | *Anomala* sp4 | 0.76 | PT | *Harpalus* sp2 | 0.91 |
| **LT 500** | *Anomala* sp6 | 0.57 | PT | *Hiletus* sp1 | 0.44 |
| **LT 500** | *Apogonia* sp3 | 0.79 | PT | *Inopeplus* sp1 | 0.63 |
| **LT 500** | *Apogonia* sp5 | 0.79 | PT | *Lebia* sp1 | 0.57 |
| **LT 500** | *Cicindela* sp2 | 0.79 | PT | *Omonadus* sp1 | 0.52 |
| **LT 500** | Mordellidae A | 0.80 | PT | *Pentagonica* sp1 | 0.54 |
| **LT 500** | Scarabeidae M | 0.80 | PT | Staphylinidae M | 0.63 |
| **LT 500** | Scarabeidae P | 0.58 | PT 500 | *Pterostichus* sp2 | 0.87 |
| **LT 500–1000** | *Epepeotes lateralis* | 0.52 | PT 500–1500 | *Spinolyprops* A | 0.53 |
| **LT 1000** | *Apogonia* sp1 | 0.90 | PT 500–1500 | Staphylinidae C | 0.61 |
| **LT 1000** | *Cicindela* sp1 | 0.69 | PT 1000 | *Actiastes* sp1 | 0.71 |
| **LT 1000–1500** | *Anomala* sp1 | 0.94 | PT 1000 | *Anotylus* sp2 | 0.86 |
| **LT 1000–1500** | *Apogonia* sp2 | 0.96 | PT 1000 | *Bledius* sp1 | 0.78 |
| **LT 1500** | Lampyridae E | 0.70 | PT 1000 | *Lispinus* sp1 | 0.84 |
| **LT 1500** | *Luciola* sp1 | 0.71 | PT 1000 | *Orphnebius* sp1 | 0.84 |
| **LT 1500–1800** | *Altica* sp1 | 0.69 | PT 1000 | *Orphnebius* sp2 | 0.92 |
| **LT 1800** | *Cleorina* sp | 0.71 | PT 1000 | Passalidae A | 0.44 |
| **LT 1800** | *Hoplocerambyx spinicornis* | 0.87 | PT 1000 | *Pityogenes* sp1 | 0.74 |
| **LT 1800** | *Hydrovatus enigmaticus* | 0.97 | PT 1000 | *Sunius* sp1 | 0.71 |
| **LT 1800** | *Illeis* sp2 | 0.70 | PT 1000–1800 | *Oxylatus* sp1 | 0.57 |
| **MA 500** | Aleocharinae sp1 | 0.84 | PT 1500 | *Hiletus* sp2 | 0.58 |
| **MA 500** | *Anotylus* sp1 | 0.94 | PT 1500 | *Pterostichus* sp1 | 0.79 |
| **MA 500** | *Aphthona* sp1 | 0.98 | PT 1500–1800 | *Paederus* sp1 | 0.69 |
| **MA 500** | *Brachypeplus* sp1 | 0.92 | PT 1800 | *Lebia* sp2 | 0.57 |
| **MA 500** | *Brachypeplus* sp3 | 0.98 | PT 1800 | *Pterostichus* sp3 | 0.79 |
| **MA 500** | *Bradymerus* sp2 | 0.99 | | | |
| **MA 500** | *Epuraea* sp1 | 0.70 | **Multiple-method indicators** | | |
| **MA 500** | Galerucinae sp1 | 0.97 | MA+LT 500 | *Mulsanteus* sp1 | 0.60 |
| **MA 500** | *Ischnosoma* sp1 | 0.95 | MA+LT 500 | Paederinae sp3 | 0.68 |
| **MA 500** | *Lymantor* sp1 | 0.94 | MA+LT 1000 | *Sarmydus* sp1 | 0.71 |
| **MA 500** | *Lymantor* sp2 | 0.91 | MA+LT 1000–1500 | *Strotocera* sp1 | 0.71 |
| **MA 500** | *Lymantor* sp3 | 0.94 | MA+LT 1500 | Alticinae sp2 | 0.69 |
| **MA 500** | *Sinoxylon* sp1 | 0.99 | MA+LT 1500 | *Anisandrus* sp1 | 0.68 |
| **MA 500** | *Xyleborus* sp1 | 0.98 | MA+LT 1500 | *Brachypeplus* sp2 | 0.65 |
| **MA 500–1000** | Aleocharinae sp2 | 0.58 | MA+LT 1500 | *Colaspoma* sp2 | 0.71 |
| **MA 1000** | *Anomala* sp3 | 0.82 | MA+LT 1500 | Curculionidae A | 0.70 |
| **MA 1000–1500** | Alticinae sp1 | 0.70 | MA+LT 1500 | Meloidae A | 0.71 |
| **MA 1500** | Paederinae sp2 | 0.66 | MA+LT 1500 | *Nisotra* sp2 | 0.71 |
| **MA 1500–1800** | *Xyleborus* sp2 | 0.71 | MA+LT 1500 | *Pityogenes* sp2 | 0.48 |
| **MA 1800** | *Anomala* sp5 | 0.66 | MA+PT 500 | Aleocharinae sp3 | 0.62 |
| **MA 1800** | *Apogonia* sp4 | 0.69 | | | |
| **MA 1800** | *Lymantor* sp4 | 0.90 | | | |
| **MA 1800** | *Xylothrips* sp1 | 0.68 | | | |

LT = indicator of light traps; MA = indicator of Malaise traps; PT = indicator of pitfall traps; different Altitudes or ranges are shown with numbers (500, 1,000, 1,500, 1,800 m a. s. l.). Column IndVal shows indicator value for each taxon.

variation. Moreover, and only at Fraser's Hill, Malaise traps distinguished 1,800 m from the lower altitudes, and pitfall traps distinguished 500, 1,000, and 1,500–1,800 m.

## Indicator species for sampling methods and altitudes

Pitfall traps produced eight indicator species that were common at all altitudes, whereas the other sampling methods did not produce such altitude generalists (for identities of these species, see Table 4). This finding is in line with our rank-abundance plots (Fig 4). Moreover, all sampling methods produced at least some indicators of low (500–1,000 m), intermediate (1,000–1,500 m), and/or high altitudes (1,500–1,800 m), suggesting potential for each method to reflect altitude and land use. Important from a conservation perspective, we found nine significant indicators of 1,800 m (four using light, four using Malaise, and one using pitfall traps). Species adapted to high altitudes face risks posed by intensifying land use, as exemplified by pollinators [62, 63], but with predicted climate warming, poorly-dispersing species occupying mountain tops have limited chances to spread (e.g., [64]). These sorts of phenomena may also act in concert, which would warrant follow-ups of sampling in the studied mountains.

In our study area, the two mountains hosted very different beetle communities, which perhaps partly reflects different intensities of land-use. The studied sampling methods differ in costs, required labor, relative capturing efficiency, and captured species pool. The differences reported here suggest that the use of more than one location and several sampling methods are desirable in environmental assessments [12, 65, 66].

## Research caveats

Taxonomic constraints may have affected our results to some extent, as most taxa had been identified to family or genus levels only. Thus, two individuals of a given taxon being captured using two methods or at different altitudes might in reality be two different species. However, this possibility rather masks than exaggerates the true community responses. We are therefore confident that the sampling methods and altitudes truly differed in beetle communities, but the differences would have been more pronounced had we been able to identify everything to species.

## Conclusions

The location of the present study, Malaysia, belongs to the global biodiversity hotspot of Sundaland, yet little is known about the invertebrate diversity of its mountains [67, 68]. Results of our study are applicable to tropical species conservation: they provide evidence for adaptations of many species to particular altitudes and, more importantly, differences in beetle samples between collecting methods. The light and Malaise traps showed little difference in terms of species composition but differed remarkably from pitfall-trap data. The three sampling methods also reflected the altitudinal gradient in different ways, and many species were trap-type specific. Clearly, caution is required while interpreting environmental impact on biodiversity based on one sampling method only. Whenever possible, we strongly recommend multiple collecting methods in environmental impact assessments on biodiversity. This is particularly important in land-use or political decision making, which should ideally be based on a holistic picture of biodiversity to avoid unwanted species losses or changes in ecosystem functioning.

## Supporting information

**S1 Table. GLMM outputs for data collected using different sampling methods.**
(DOCX)

**S2 Table. Wilk-Shapiro test for GLMM residuals plotted in S1 Fig.**
(DOCX)

**S3 Table. Wilk-Shapiro (W) and Breusch-Pagan (BP; df = 4) tests for ANOVA residuals; model Mountain + Altitude.**
(DOCX)

**S4 Table. Robust LMM output for the number of species, rarefied richness standardized to 5, 10 or 20 individuals and coverage-based asymptotic richness for three trapping methods.**
(DOCX)

**S5 Table. Species captured using three collecting methods at the two study mountains; numbers are total catches.**
(DOCX)

**S6 Table. Twenty most abundantly captured species in our data, sorted according to trap-type specific sample sizes; n indicates number of individuals in the data.** Light gray: species shared between light and Malaise traps; dark gray: species shared with pitfall, light and Malaise traps.
(DOCX)

**S1 Fig. Q-Q plots for GLMM residuals, shown separately for three sampling methods (compare S1 Table).** Different rows show plots for the number of species, rarefaction standardized richness based on 5, 10 or 20 individuals, and coverage-based (CB) rarefied asymptotic richness. Solid line shows perfect fit, dash lines show 95% confidence intervals. Ideally, all sample residuals should fall between confidence intervals.
(DOCX)

## Acknowledgments

The authors would like to acknowledge the anonymous reviewers of the manuscript which helped to improve the article to a great deal. We thank Mohd Shukri Mohd Sabri and Davin-dram A/L Rajendram for assistance in beetle sampling. We also acknowledge the Universiti Kebangsaan Malaysia and Forestry Department of Malaysia for supporting this study, and Marcelo Bruno Pessôa and an anonymous reviewer for constructive comments.

## Author Contributions

**Conceptualization:** Fauziah Abdullah.

**Data curation:** Muneeb M. Musthafa.

**Formal analysis:** Matti J. Koivula.

**Funding acquisition:** Fauziah Abdullah.

**Investigation:** Muneeb M. Musthafa.

**Methodology:** Muneeb M. Musthafa, Fauziah Abdullah.

**Software:** Matti J. Koivula.

**Writing – original draft:** Muneeb M. Musthafa.

**Writing – review & editing:** Matti J. Koivula.

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
