## [Decision Letter · Decision Letter 0]

16 Mar 2021

PONE-D-21-03456

Community composition and indicator functioning of beetles across environmental gradients in Peninsular Malaysia

PLOS ONE

Dear Dr. Musthafa,

Thank you for submitting your manuscript to PLOS ONE. After careful consideration, we feel that it has merit but does not fully meet PLOS ONE’s publication criteria as it currently stands. Therefore, we invite you to submit a revised version of the manuscript that addresses the points raised during the review process.

We look forward to receiving your revised manuscript.

Kind regards,

Daniel de Paiva Silva, Ph.D.

Academic Editor

PLOS ONE

Journal Requirements:

4. We note that Figure 1 in your submission contain map images which may be copyrighted. All PLOS content is published under the Creative Commons Attribution License (CC BY 4.0), which means that the manuscript, images, and Supporting Information files will be freely available online, and any third party is permitted to access, download, copy, distribute, and use these materials in any way, even commercially, with proper attribution. For these reasons, we cannot publish previously copyrighted maps or satellite images created using proprietary data, such as Google software (Google Maps, Street View, and Earth). For more information, see our copyright guidelines: http://journals.plos.org/plosone/s/licenses-and-copyright.

(1) You may seek permission from the original copyright holder of Figure 1 to publish the content specifically under the CC BY 4.0 license. 

Additional Editor Comments:

Dear Musthafa et al.,

after the full consideration of your manuscript by two independent reviewers, I believe your manuscript could be accepted for publication in PLoS One after you perform a major review to your text. Both reviewers raised significant changes to be made in your text, that I believe will significantly improve its final version of the text. Considering the current COVID-19 pandemic scenario, I believe that a three months period will be more than enough for you to perform the necessary changes to your manuscript. Therefore, please resubmit your text up to June 15th 2021. Please, do not forget to prepare a rebuttal letter, informing the reviewers the changes you agreed on and those you disagreed. In this rebuttal letter, you will explain each and every improvement you did, considering the reviewers' suggestions and also the reasons why you did not do some of the changes (in case it happens).

If you are able to finish the new version of the text before the due date specified above, do not hesitate to resubmit earlier. Still, in case the three months period is not enough. Please let me or PLoS One's staff know, so we can find a better due date that fits your needs. Finally, in case you have any other doubts, please contact me without hesitation.

Sincerely,

Daniel Silva

Reviewers' comments:

Reviewer's Responses to Questions

**Comments to the Author**

1. Is the manuscript technically sound, and do the data support the conclusions?

Reviewer #1: Yes

Reviewer #2: Yes

2. Has the statistical analysis been performed appropriately and rigorously? 

Reviewer #1: Yes

Reviewer #2: No

3. Have the authors made all data underlying the findings in their manuscript fully available?

Reviewer #1: No

Reviewer #2: No

4. Is the manuscript presented in an intelligible fashion and written in standard English?

Reviewer #1: Yes

Reviewer #2: No

5. Review Comments to the Author

Reviewer #1: • TITLE

o The title does not reflect the main idea of the research. In my view, the main objective is to test different trap methodologies.

o The functioning and environmental gradient do not add up to the work. The study is only with altitudes gradients.

• INTRODUCTION

o I missed in the introduction a better explanation about the methods. Why do you expect that the traps will have different outcomes? How beetle biology may affect these differences?

o The Climate change sentence is lost in the text, and don’t add up to your objective, you can use it in the discussion and conclusion part, but it does not make any difference in the introduction.

o need to make it clearer in the introduction that the goal in the introduction is to compare the methods.

• METHODS

o Could you show the species accumulation curve as supplemental information, with the hole composition as well.

o What is the letters A, B, C, D, E, F, G, H? => If they are Stands you have to explain better.

o Get the GLMM model more explicit.

o Figure 2 has some problems. Standardize the axes so you can better compare the results. It would be good to add a table with richness values by the mountain, altitude, and trap.

o Figure 3 identifies the species, by name in the legends.

• RESULTS

o NMDS- The 1800 altitude separation becomes evident only in the permanova.

• DISCUSSION

o Discuss more the implications of the differences of the methods.

o What is the bias towards decision-making using only a method and possible vantages to use more than one method?

o If you are talking about trap efficiency, since pitfall with a lesser richness capture differences in all elevations, and combinations I would assume this is the most efficient method (if you emphasize more the different biologies related to each trap you can sell better the idea of complementary survey methods).

o Line 263: wind direction is duplicated.

o

• CONCLUSION

o The recommendation part needs to be better, talk about the biology of beetles and how this would affect the methodology choice of your research. Remember that methodology should adequate to the question made.

o Improve the text so it is not a resume of the paper, you can make a more thorough recommendation, a guide for decision making. As it is now the recommendation is too subtle.

• FIGURES

o Figura 1 – Put the names of the mountains in the first image. Identify the images A, B e C.

o Figure 4: Use different colors for elevation and different symbols for mountains.

• TABLES

o Why only 500m in this table? Explain this better in the methods.

• GENERAL

o Text organization of methods and results is really good and with easy understanding.

Reviewer #2: The paper titled “Community composition and indicator functioning of beetles across environmental gradients in Peninsular Malaysia” has important contributions for studies on environmental gradients. The comparison of methods is interesting and this is the main objective of the work. The authors did a huge sampling effort and made a good analysis proposal. However, the paper has some issues and some improvements are needed. Most issues are related to the organization of ideas. Also, the paper has some writing issues. The English is good, but they need to improve the writing. I made several comments along the text trying to highlight some critical points, but I recommend a full text revision. Two specific things that were hard to understand were the sampling design and the analysis descriptions, for which they need to provide more information. Moreover, they need to improve the model diagnostics. I made some suggestions in the Material and Methods. The figures, specially the map, need improvements. Therefore, I recommend major revision and suggest that the authors incorporate the suggestions made throughout the text.

6. PLOS authors have the option to publish the peer review history of their article (what does this mean?). If published, this will include your full peer review and any attached files.

Reviewer #1: **Yes: **Marcelo Bruno Pessôa

Reviewer #2: No

---

## [Author Response · Author response to Decision Letter 0]

6 May 2021

• TITLE

o The title does not reflect the main idea of the research. In my view, the main objective is to test different trap methodologies.

o The functioning and environmental gradient do not add up to the work. The study is only with altitudes gradients.

Response: this is true. We rewrote the title accordingly.

• INTRODUCTION

o I missed in the introduction a better explanation about the methods. Why do you expect that the traps will have different outcomes? How beetle biology may affect these differences?

o The Climate change sentence is lost in the text, and don’t add up to your objective, you can use it in the discussion and conclusion part, but it does not make any difference in the introduction.

o need to make it clearer in the introduction that the goal in the introduction is to compare the methods.

Response:

1. We dedicated one paragraph to method outcomes. Beetle biology indeed impacts this, but as with many tropical regions, including the studied one, their fauna, species and ecologies are extremely poorly understood. Hence we can mostly only speculate about species biology here.

2. We see the reviewer's point here, and moved climate texts to Discussion.

3. We clarified the end of introduction to underscore that the aim is to compare methods.

• METHODS

o Could you show the species accumulation curve as supplemental information, with the hole composition as well.

Response:

1. We added rarefaction curves for each trap type and pooled sample (new Fig. 2). This shows that only pitfall trap catches saturated, whereas light and Malaise traps continued to trap new species, which was reflected in the total sample as well. We omitted the initial accumulation analysis as it contained a calculus error.

2. We also omitted the PERMANOVA approach as it was done with the pooled sample and intended to compare elevations only, whereas our aim was, following the reviewers’ advice, to compare methods (and elevation is a "side product"). Additionally, this analysis did not produce anything new as compared to the other analyses: elevations are compared in much more detailed ways using GLMM, NMDS, MRT and IndVal. So we prefer to leave out also this whole-data curve.

o What is the letters A, B, C, D, E, F, G, H? => If they are Stands you have to explain better.

Response: Thanks for pointing out this lack of clarity. These are indeed study stands, as we now say in the figure text (stands denoted with letters A-J), and cite Table 1 where these are shown.

o Get the GLMM model more explicit.

Response: The full GLMM output is shown in the Supplementary materials. We believe that an average reader will appreciate a simple, at-glance presentation of initially quite diverse analysis outputs. However we are ready to present the big GLMM tables in the manuscript (and not in Supplementary) if the reviewer and editors require that.

o Figure 2 has some problems. Standardize the axes so you can better compare the results. It would be good to add a table with richness values by the mountain, altitude, and trap.

Response:

1. While axis standardization is generally a valid point, in our case it would lead to many of the pitfall-trap columns to become too short to be distinguishable. Therefore, we did not change the figure but added a note to figure legend that the X axis scales are different.

2. The suggested table would be our initial data set, which we will provide in the journal's open access database if our manuscript gets accepted.

o Figure 3 identifies the species, by name in the legends.

Response: we assume that the reviewer would like us to show the species identities for each column. We added an Appendix table that shows these species in rank order per method, with shared species highlighted.

• RESULTS

o NMDS- The 1800 altitude separation becomes evident only in the permanova.

Response: it is indeed common that differences are not that clear in a 2-dimensional graph, whereas permanova considers also other dimensions in the data. The separation is very clear in Table 3, and particularly regarding light traps, the result is extremely clear even without these, as the 1800-m and the other scores do not overlap (Fig. 4). We believe that, after rearranging (and partly rewriting) the methods and statistical outputs, the impact of elevation is now considerably more clearly demonstrated.

• DISCUSSION

o Discuss more the implications of the differences of the methods.

Response: We added notes on practical aspects of using these methods to Discussion, and organized the text better to highlight the use. However we need to underscore that quite little is known about the captured species, which limits conservation implications. We highlight this too in Discussion.

o What is the bias towards decision-making using only a method and possible vantages to use more than one method?

Response: this is an important point. We clarify the end of Discussion as follows: "This is an important aspect as land-use or political decision making should ideally be based on as holistic picture as possible about biodiversity to avoid species losses or unwanted changes in ecosystem functioning."

o If you are talking about trap efficiency, since pitfall with a lesser richness capture differences in all elevations, and combinations I would assume this is the most efficient method (if you emphasize more the different biologies related to each trap you can sell better the idea of complementary survey methods).

Response: this may indeed be the case, and/or many species captured using pitfall traps may be too generalistic in their life histories that they may not be that relevant for conservation, after all, which would make pitfall traps less optimal for studying elevational or climate gradients. But of course, to really determine conservation relevance would require knowledge about species identities, and population sizes, locations and long-term trends. But the main reason for recommending the use of multiple methods is based on complementarity, as explained above.

Few sentences on species ecology has been added with the details of generalists and specialist species. 

o Line 263: wind direction is duplicated.

Response: thanks for pointing this out. The other was deleted.

• CONCLUSION

o The recommendation part needs to be better, talk about the biology of beetles and how this would affect the methodology choice of your research. Remember that methodology should adequate to the question made.

Response: this is a valid criticism. We added discussion about links between species life histories and collecting methods (see Conclusions in the end of Discussion).

o Improve the text so it is not a resume of the paper, you can make a more thorough recommendation, a guide for decision making. As it is now the recommendation is too subtle.

Response: this is valid criticism. We tried to improve this with a stronger statement. See our proposed addition to your comment regarding decision making.

• FIGURES

o Figura 1 – Put the names of the mountains in the first image. Identify the images A, B e C.

Done accordingly and new figure has been added. 

o Figure 4: Use different colors for elevation and different symbols for mountains.

Response: we prefer not to add more information to this already quite complex graph, plus mountains were not really of interest as our focus was on methods and how they reflect altitudes.

• TABLES

o Why only 500m in this table? Explain this better in the methods.

This shows a distinction between 500 m and the other Altitudes. The other altitudes are compared to the rest below the 500 row. We clarify this in Methods.

• GENERAL

o Text organization of methods and results is really good and with easy understanding.

Response: thank you for this comment.

Reviewer #2

The paper titled “Community composition and indicator functioning of beetles across environmental gradients in Peninsular Malaysia” has important contributions for studies on environmental gradients. The comparison of methods is interesting and this is the main objective of the work. The authors did a huge sampling effort and made a good analysis proposal. However, the paper has some issues and some improvements are needed. Most issues are related to the organization of ideas. Also, the paper has some writing issues. The English is good, but they need to improve the writing. I made several comments along the text trying to highlight some critical points, but I recommend a full text revision. Two specific things that were hard to understand were the sampling design and the analysis descriptions, for which they need to provide more information. Moreover, they need to improve the model diagnostics. I made some suggestions in the Material and Methods. The figures, specially the map, need improvements. Therefore, I recommend major revision and suggest that the authors incorporate the suggestions made throughout the text.

Response: thank you for these comments, which are all important. We attempted to clarify the text and reorganize the structure according to both referees’ comments (see also responses to referee 1). We went through all comments placed in the manuscript pdf, thank you for such thorough job. Specifically, we made the following major changes (linguistic or clarity-related suggestions were mostly accepted as such).

Abstract

The indicator hypothesis was removed while strengthening the focus on methods (see reviewer #1). We now use altitude, not elevation, throughout the text, and name the compared traps in Abstract. PERMANOVA was removed (see reviewer #1). From results in Abstract, we removed analysis names and clarified the tested characteristics of beetle assemblages.

Introduction

Thank you for pointing out Rahbek et al. 2019, which we now cite in Introduction.

We rewrote the global hotspot sentence as suggested.

Request for clarifying altitude and climate: this part was moved to Discussion, where we rewrote it to suggest that changes in temperature, etc. with altitude might have some similar effects as global climate change, giving a reason to study mountain slopes.

The messy paragraph 2: we agree in that we had packed too much scattered information into this. We therefore rewrote the paragraph to be better linked with our aim, which is methodological comparison, whereas altitude is a side effect.

We kept the word "largely" as the determination concerns most but not all responses. However, we went through the whole text to omit similar, unwarranted words. We too think they should be avoided.

The "three mentioned trap types": we now mention only the three studied trap types in the previous paragraph. For some reason the initial text included flight intercept traps, which is now deleted.

We moved the sentences that the reviewer indicated to appropriate sections.

Our intention is not to assess sampling efficiency but rather the consistency of methods in reflecting altitude. Hence, we did not change the initial sentence.

With these edits we believe that the text logic is now much clearer.

Methods

It is an excellent idea to reorganize this section according to study questions. So we rearranged Methods, Results and Discussion. Also, as we noted that the analyses actually describe four (instead of three) viewpoints, we adjusted the research questions accordingly.

We adopted all linguistic suggestions by the reviewer.

Regarding criticism on the trap section containing messy details, we reorganized the presentation as follows: (1) general design (mountains, altitudes), (2) trap types and their numbers, (3) how traps were placed per altitude, and (4) what were the trapping periods. We clarified the trapping-period length (one day per month), samples from different periods were pooled, and that each trap, at least 150 m from the nearest other trap, is considered a replicate in analysis. We also clarified that we had two mountains, each with four altitudes, and each such combination had 2 light, 2 Malaise and 5 (groups of 5) pitfall traps, making up initially 72 samples.

We restructured and clarified the trapping protocol to follow the reviewer's advice.

We ran all statistical analyses using 69 samples without any exceptions, as we say in the beginning of statistics descriptions. We also refer to Table 1 where these samples are linked with the ten stands shown in Fig. 1. Rarefaction is a standard method nowadays, and was calculated for each of the 69 samples (note: no exceptions).

GLMM: we see the reviewer's point in adding more explanatory variables. It is a common problem in mountain research that many things change in concert. Regrettably we do not have additional data, as no forest inventory, for example, was done. It is also worth noting that more complex models result in poorer generalizations. Nevertheless, we clearly acknowledge this shortcoming and propose alternative or, rather, complementary explanations in Discussion.

Thank you for suggesting an alternative GLMM tool. However, our approach appeared robust with alternative approaches, and we also used a well-established assessment of data normality and model robustness, so we did not change this analysis.

Turnover: The important thing is that light and Malaise traps shared about half of their abundant species, whereas pitfall traps produced rather unique data. It also captures clear changes with elevation, and is easy to understand. However, statistical power for this aspect comes from permanova for methods and elevations. We reorganized the text to better underline these results, and link them clearer to the turnover analysis.

Results:

The initial accumulation calculi had an error, plus our intention is really to compare trap types, so we recalculated accumulation curves for each trap type for each of the two mountains. These are now shown as a figure, Appendix 1.

Assemblage structure and turnover: we have now clarified and restructured Methods and Results as there was apparently some lack of clarity. The statements regarding similarity are partly based on the rank-abundance plots, but partly also on permutational multivariate ANOVA (see rewritten Results and Table 3).

Discussion:

Abiotic versus biotic factors affecting species: we do not have data for these; therefore we say "might". At a general level, of ecology, biotic and abiotic factors impact species.

The referee has often commented our speculations and suggested us to provide data. We would be happy to do so, had we some. So neither do we have such data nor large libraries on the ecology of the sampled species, which we highlight in Discussion. For these reasons we express us cautiously. But we believe that even speculative sections may function as seeds for future research.

We did not use the general forest type in analyses because technically it would be the same thing as using altitude as a factor. But of course, as we say clearly in Discussion and in our above comments, altitude probably reflects many things and not just height from sea level; one is habitat type for sure, others are temperature, windiness, etc.

We now refer to Table 4 to show which species indicated what, according to the referee's suggestion.

We moved the statement about altitudinal variation related to mountain and method to the beginning of the paragraph. Thanks for pointing out one of the key findings!

We see the point that our discussion regarding taxonomic level may sound repeating. However, we need to make a point here: specimen of genus A at two different locations may in reality be two different species, whereas if we consider them to be the same (as technically we do so in our analysis), we will probably miss some true location effects.

---

## [Decision Letter · Decision Letter 1]

5 Aug 2021

PONE-D-21-03456R1

The functioning of different beetle (Coleoptera) sampling methods across altitudinal gradients in Peninsular Malaysia

PLOS ONE

Dear Dr. Musthafa,

Thank you for submitting your manuscript to PLOS ONE. After careful consideration, we feel that it has merit but does not fully meet PLOS ONE’s publication criteria as it currently stands. Therefore, we invite you to submit a revised version of the manuscript that addresses the points raised during the review process.

Please submit your revised manuscript by November 5th, 2021 If you will need more time than this to complete your revisions, please reply to this message or contact the journal office at plosone@plos.org. Please include the following items when submitting your revised manuscript:

We look forward to receiving your revised manuscript.

Kind regards,

Daniel de Paiva Silva, Ph.D.

Academic Editor

PLOS ONE

Journal Requirements:

Additional Editor Comments (if provided):

Dear Musthafa et al,

After the second round of reviews, I need to say that I agree with reviewer #2, who indicated that several improvements are still necessary to make the study acceptable for publication.

Considering the extent of improvements that are needed to be done, I will grant you with a three-months period to resubmit. Please prepare a point-by-point rebuttal letter, explaining all the improvements that were done, and all of those which you did not agreed with. Please resubmit by November 5th, 2021. In case you need more time, please let me know. Nonetheless, do not hesitate to resubmit earlier if you are able to.

Sincerely,

Daniel Silva, Ph.D.

Reviewers' comments:

Reviewer's Responses to Questions

**Comments to the Author**

1. If the authors have adequately addressed your comments raised in a previous round of review and you feel that this manuscript is now acceptable for publication, you may indicate that here to bypass the “Comments to the Author” section, enter your conflict of interest statement in the “Confidential to Editor” section, and submit your "Accept" recommendation.

Reviewer #2: (No Response)

Reviewer #3: All comments have been addressed

2. Is the manuscript technically sound, and do the data support the conclusions?

Reviewer #2: Partly

Reviewer #3: Yes

3. Has the statistical analysis been performed appropriately and rigorously? 

Reviewer #2: No

Reviewer #3: Yes

4. Have the authors made all data underlying the findings in their manuscript fully available?

Reviewer #2: Yes

Reviewer #3: Yes

5. Is the manuscript presented in an intelligible fashion and written in standard English?

Reviewer #2: No

Reviewer #3: Yes

6. Review Comments to the Author

Reviewer #2: Dear Authors,

I appreciate the opportunity to review your paper again. I really like of your manuscript. Your main find is a great contribution for future environment studies. However, I did not see many of the previous suggestions in the paper. I said "suggestion", because I believe I am not in position to say what you have to do. I can just point out the weaknesses of your manuscript and suggest solutions. If you have a better solution for the problem, this is up to you to do. Thus, there are many weaknesses points in your manuscript that I saw again in this version, mainly related to:

- Map - see the comments in the text;

- Analysis - You must appoint how did you diagnostic the model? How did you find the best distribution? How did you support the normality of your data? At least, you must diagnose the normality of residuals and homogeneity of variance. I do not recommend transform your data. You must try to fit your data in some distribution (there are many);

The information of your manuscript are confuse. You must find a better way to organize and describe your MM and results. I did many suggestions about this in the previous review. I saw many changes in your discussion, but it seems very speculative (see the comments in the text). I suggest a double-check before to submit the manuscript again, there are many problems of Supplementary, Tables, Figures, commas and some sentences are without context.

Best regards!

Reviewer #3: This article is a welcome contribution to improve the efficiency of sampling methods in tropical environments. The positive outcome of the study, but also its limitations, are properly assessed.

7. PLOS authors have the option to publish the peer review history of their article (what does this mean?). If published, this will include your full peer review and any attached files.

Reviewer #2: No

Reviewer #3: No

---

## [Author Response · Author response to Decision Letter 1]

28 Oct 2021

• If you want to keep this format to describe your results, I recommend rewrite something like this: "Our finds were..."

 This sentence had been rewritten as suggested

• …. only light traps...

 Corrected as suggested 

• It seems a little vague, please be more specific describing your results in the abstract. It is your time to show a compilation of your paper. Which is considerable? Which is less so?

 This sentence is sharpened as follows: “Only light traps (but not the other trap types) distinguished high from middle or 

 low altitudes in terms of beetle richness and assemblage composition.”

• Please, rewrite! It seems little confuse this sentence as well as I specified in the first review: 

Community composition – species, their relative shares, and interactions (biotic responses) –largely determines the general biodiversity response to environmental variation. The Climate change sentence is lost in the text, and don’t add up to your objective, you can use it in the discussion and conclusion part, but it does not make any difference in the introduction.

 This sentence was indeed unclearly written so we rewrote it as follows: “Community composition is largely determined by 

 the relative abundances of, and interactions among, its species, which contribute to the general biodiversity response to 

 environmental variation.” We also omitted most of the climate notes from the manuscript as the reviewer is right in that 

 we did not directly assess climatic issues.

• Please, rewrite this sentence. 

 I suggest: "The objective of the present study is evalute the consistency of light, Malaise and pitfall trap samples in 

 relation to the altitudinal gradient".

 Following the reviewers advise, we modified the sentence as: “The objective of the present study is to evaluate the 

 consistency of light, Malaise and pitfall trap samples in reflecting an altitudinal gradient in tropical Malaysia.”

• Exclude this comma: We removed the extra comma. 

• Please, see the comment in the map: 

 Please, try to improve this map. Your map has several problems. I can't see which group of points represent each altitude 

 (maybe is the map resolution, but it is better represent this with some color in the legend). Some points are overlap 

 (there are many function in QGIS that you can correct this problem - for example see in points 

 proprieties/Labels/Callouts). You need to use a better layer or some shapefile, add maps coordinates, add scale, add the 

 north arrow and structure better the overview. The previous map was better than this map.

• I still don't understand how did you arrange 25 pitfall traps in four altitudes. The map show to me only 20 pitfall traps. 

 Please, check it.

 Thank you for pointing out an error in the text! We had 40 pitfall traps so the text has been clarified as follows: “We 

 sampled beetles using two Malaise traps, two light traps, and 10 (5 groups of 2) pitfall traps at each altitude (500 m, 

 1,000 m, 1,500 m, and 1,800 m a.s.l.) at both mountain slopes.”

• Why 40 pitfall traps if you describe above 25 pitfall traps in each mountain? Please, check it: 

 See previous point: we had 40 traps (5 sets of 2 locations at 4 elevations)

• I'm still not convinced with your analysis. You assumed Gaussian distribution, because you evaluated your data using only 

 Q-Q plots and transform your data. I don't recommend transform your data. It is better to find a better distribution that 

 fits your data. You should use some analysis which you can test the normality such as shapiro test. Quasi-poisson is not a 

 distribution. You use Quasi-poisson if you found overdispersion in Poisson model. Did you evaluate your model with 

 Poisson? There is overdispersion?

 Did you perform some diagnostics after to run your model? Did you test the normality of residuals and homogeneity of 

 variance? Which R package did you perform your analysis?

 Please, rewrite your data analysis and check all these things.

 We followed the reviewer's advice by adding assessments of GLMM residual normality and heteroskedasticity. Moreover, 

 as these and the QQ plots sometimes suggested slight departures from normality, we reran the models using a novel 

 “robust LMM” approach, which is tailored for data with outliers while being technically quite similar to lme4. As these 

 reruns produced similar results to our initial GLMM, we conclude that the initial GLMM produced reliable results, so we 

 therefore maintain the initial GLMM results in the main document (but present the robust LMM results in the 

 Supplementary materials). We think it is also important to realize that the QQ plots in Supplementary files measure 

 residual normality, which is a commonly used way to assess the appropriateness of GLMM, and that none of the presented 

 analyses are or were done with transformed data (we used transformed data in some trials in the supplementary files of 

 the previous manuscript version, which are now not presented or referred to). We were apparently not very clear with 

 these things earlier on. We thus removed the various GLMM trials from the previous version of Supplementary files and 

 now show the normality, heteroskedasticity and robust LMM results (for comparisons with the initial GLMM) in the 

 Supplementary files.

 The rest of the information can be found in the manuscript. 

• How did you analyze the species accumulation curves? Which package? Please, describe this in MM: 

 We restructured Material and Methods to clarify these. (They were initially in the end of Material and Methods.) We now 

 mention each package in text sections where the analysis in question is described.

• I didn't see any significant improvement in relation to the previous version. Please, detail more your finds.

 We have now restructured and rewritten sections in Results to improve the text flow. We are unsure what the reviewer 

 means with detailing findings, but hope that the rewritten parts better fulfil this request too.

• I didn't find this Fig: This figure is referred to as Supplementary Table S6, as it is really a table not figure. Apologies for 

 the lack of clarity here!

• This seems very speculative. Please, remove: Removed as suggested 

• Again, this seems very speculative. It is obvious that there are climact changes along the altitudinal gradient, but you 

 don't have this data. You don't test this data with climact factors. Many of your explanations are related with climatic 

 factors. So, why you didn't test the climatic factors? I suggest do less emphasis to climact factors, trying to explain your 

 data and focus in your finds: 

 We agree with the reviewer in that we did not really deal with climate, but many things that change in concert with 

 altitude. This part too was removed as suggested.

• So, you can't discuss a half of what you discussed in the previous topic: We agree with the reviewer in that we did not 

 really deal with climate, but many things that change in concert with altitude. This part too was removed as suggested.

• The reader will judge this. Please, remove: We see the point here and removed the section as suggested 

• Whenever possible, we recommend the use of multiple collecting methods in environmental impact assessments on 

 biodiversity: We rewrote the recommendation as suggested by the reviewer.

• Figure 1 – Put the names of the mountains in the first image. Identify the images A, B e C: Figure 1 has been re-drawn with all the details.

---

## [Decision Letter · Decision Letter 2]

27 Dec 2021

PONE-D-21-03456R2The functioning of different beetle (Coleoptera) sampling methods across altitudinal gradients in Peninsular MalaysiaPLOS ONE

Dear Dr. Musthafa,

Thank you for submitting your manuscript to PLOS ONE. After careful consideration, we feel that it has merit but does not fully meet PLOS ONE’s publication criteria as it currently stands. Therefore, we invite you to submit a revised version of the manuscript that addresses the points raised during the review process.

We look forward to receiving your revised manuscript.

Kind regards,

Daniel de Paiva Silva, Ph.D.

Academic Editor

PLOS ONE

Additional Editor Comments (if provided):

Dear Dr. Musthafa,

After a new review round, both reviewers raised a series of issues that need to be taken care of before publication. Considering the amount of changes you need to prepare, I will provide you a major review to be submitted up to Feb 10, 2022. When you prepare the the new version of your study, please do not forget to prepare a rebuttal letter, informing of all the improvements you did to your study. Please do not hesitate to contact me in case you need more time to review your MS, still, if you are able to resubmit earlier, please do.

Sincerely and Happy 2022,

Daniel Silva, PhD

Reviewers' comments:

Reviewer's Responses to Questions

**Comments to the Author**

1. If the authors have adequately addressed your comments raised in a previous round of review and you feel that this manuscript is now acceptable for publication, you may indicate that here to bypass the “Comments to the Author” section, enter your conflict of interest statement in the “Confidential to Editor” section, and submit your "Accept" recommendation.

Reviewer #2: (No Response)

Reviewer #4: All comments have been addressed

2. Is the manuscript technically sound, and do the data support the conclusions?

Reviewer #2: Yes

Reviewer #4: Partly

3. Has the statistical analysis been performed appropriately and rigorously? 

Reviewer #2: Yes

Reviewer #4: Yes

4. Have the authors made all data underlying the findings in their manuscript fully available?

Reviewer #2: No

Reviewer #4: Yes

5. Is the manuscript presented in an intelligible fashion and written in standard English?

Reviewer #2: Yes

Reviewer #4: Yes

6. Review Comments to the Author

Reviewer #2: Dear authors, I appreciated to see many improvements of your manuscript. Only for a few things, but serious things, I cannot make the manuscript acceptable for publication. I did some comments along the attached review, but I will make some main stands here.

1 - The Figure 1 still not good. I saw some differences from previous version, but I still cannot see the sampling points and the localities. I managed to open the sampling points using the S5 file in QGIS and it is possible to make a better map.

2 - I did not find Figures 3, 4, 5 and 6 in this version of the manuscript.

3 - In the previous review I asked to avoid talk about climate change in the discussion. This subject in your manuscript it seems speculative. You can talk about conservation issues and in the final sentence, use as example land use and global climate changes. Remember, your manuscript tested the differences among sample methods along a altitude gradient and did not test any hypothesis about climate change.

Best Wishes!

Reviewer #4: PONE-D-21-03456R2: The functioning of different beetle (Coleoptera) sampling methods across altitudinal gradients in Peninsular Malaysia

This study analyses the sampling potential of three different but a priori complementary types of sampling methods for insects, specifically the study focuses on beetles.

The authors have substantially improved the manuscript since its first version, however it still can be improved in some aspects. The conclusions obtained, from my point of view, are very obvious and not very novel. However, I highlight the importance of the work as it was carried out in a geographic area with few studies on Coleoptera biodiversity in altitudinal gradients. One of my most important criticisms is the lack of an analysis of the efficiency of the sampling by the sample coverage estimator (Chao & Jost 2012: Ecology 93(12):2533–2547) that would have allowed first to check the quality of the samplings and secondly a comparison of the richness of species between types of sampling methods much more valid. These analyses could be complementary to the GLMM since the corrected Species Richness values would be used in the latter (if this were the case).

I also consider it important to add information on the temperature and humidity gradient (at least) in the Material and methods section since part of the results and the discussion focus on the importance of the environmental gradient in a climate change scenario.

Lastly, I have observed some minor mistakes that need to be considered:

Line 31: “altitude”

Line 71: Delete “.”

Line 109: Include the colour of the pitfall traps.

7. PLOS authors have the option to publish the peer review history of their article (what does this mean?). If published, this will include your full peer review and any attached files.

Reviewer #2: No

Reviewer #4: No

---

## [Author Response · Author response to Decision Letter 2]

16 Feb 2022

Editors of PLOS ONE

Dear Sir/Madam,

We thank you for a chance for letting us to improve our manuscript “The functioning of different beetle (Coleoptera) sampling methods across altitudinal gradients in Peninsular Malaysia” and submit a revised version to be considered for publication.

We have now made the following changes according to the reviewers’ advice. If no changes were made, we justify this decision below. We hope that the manuscript now satisfies the PLOS ONE editors.

On behalf of all authors,

Muneeb M. Musthafa

Reviewer 2:

1. The Figure 1 still not good.

We have now used a freely-available terrain map as a basis for this figure, and added scale bars and compass arrows. On these maps we have plotted the locations of the ten sites per mountain (referred to as letters A-J). Additions of precise locations of every trap would have made the map unreadable (realistically, this would be a maximum of 10-15 cm wide in a journal page); however, should anyone be interested in these, coordinates are given in Supplementary materials.

2. I did not find Figures 3, 4, 5 and 6 in this version of the manuscript.

Apologies for this, we hope this was not an error of ours. We nevertheless pay attention in including all figures (1-6), tables and supplementary materials. The data are added as another supplementary file in text format.

3. In the previous review I asked to avoid talk about climate change in the discussion. This subject in your manuscript it seems speculative. You can talk about conservation issues and in the final sentence, use as example land use and global climate changes. Remember, your manuscript tested the differences among sample methods along a altitude gradient and did not test any hypothesis about climate change.

We agree with this statement and have omitted climate discussions, except for one point in Discussion where additional follow-ups are recommended (for climate reasons).

Reviewer 4:

1. I highlight the importance of the work as it was carried out in a geographic area with few studies on Coleoptera biodiversity in altitudinal gradients.

We agree with this note and highlighted this on lines 53-54.

2. One of my most important criticisms is the lack of an analysis of the efficiency of the sampling by the sample coverage estimator (Chao & Jost 2012: Ecology 93(12):2533–2547) that would have allowed first to check the quality of the samplings and secondly a comparison of the richness of species between types of sampling methods much more valid. These analyses could be complementary to the GLMM since the corrected Species Richness values would be used in the latter (if this were the case).

Thank you for this note, and pointing out this richness approach. We applied the Chao & Jost paper, used iNEXT package to calculate the coverage estimator asymptotes for each sample, and subjected these to the same GLMM approaches as for the rest of the richness estimates. The model performances are added to respective Tables and supplementary materials. Regrettably the pitfall data performed poorly due to a few apparent outliers – these could have been collected on richer than average soils, but we do not have data on this – and the robust LMM did not shed much light on this, except for confirming that pitfall data indeed largely reflected differences between mountains and altitudinal effect was much smaller.

3. I also consider it important to add information on the temperature and humidity gradient (at least) in the Material and methods section since part of the results and the discussion focus on the importance of the environmental gradient in a climate change scenario.

While we agree in that these measures are of importance or interest, the author MMM is currently preparing another manuscript on these variables. Therefore, we would not like to include these here. However, we refer to the general patterns of temperature, humidity and luminosity according to altitude (lines 87-90). Also, as the altitudinal effects generally reflected this to be quite small, we point out these generalities once again in Discussion (line 317).

4. Lastly, I have observed some minor mistakes that need to be considered

Thank you for pointing these out – these are all corrected now.

---

## [Editor Report · Decision Letter 3]

15 Mar 2022

The functioning of different beetle (Coleoptera) sampling methods across altitudinal gradients in Peninsular Malaysia

PONE-D-21-03456R3

Dear Dr. Musthafa,

We’re pleased to inform you that your manuscript has been judged scientifically suitable for publication and will be formally accepted for publication once it meets all outstanding technical requirements.

Kind regards,

Stephanie S. Romanach, Ph.D.

Academic Editor

PLOS ONE

Additional Editor Comments:

In your Acknowledgements you thank "an anonymous reviewer". For your records, PLOS has provided reviews from four peer reviewers on the first three versions (original, R1, R2) of your manuscript.

---

## [Editor Report · Acceptance letter]

21 Mar 2022

PONE-D-21-03456R3 

The functioning of different beetle (Coleoptera) sampling methods across altitudinal gradients in Peninsular Malaysia 

Dear Dr. Musthafa:

I'm pleased to inform you that your manuscript has been deemed suitable for publication in PLOS ONE. Congratulations! Your manuscript is now with our production department. 

Kind regards, 

on behalf of

Dr. Stephanie S. Romanach 

Academic Editor

PLOS ONE